# The Genus *Cetraria* s. str.—A Review of Its Botany, Phytochemistry, Traditional Uses and Pharmacology

**DOI:** 10.3390/molecules27154990

**Published:** 2022-08-05

**Authors:** Marta Sánchez, Isabel Ureña-Vacas, Elena González-Burgos, Pradeep Kumar Divakar, Maria Pilar Gómez-Serranillos

**Affiliations:** Department of Pharmacology, Pharmacognosy and Botany, Faculty of Pharmacy, Universidad Complutense de Madrid (UCM), 28040 Madrid, Spain

**Keywords:** *Cetraria*, lichens, traditional uses, pharmacology, phytochemistry, botany

## Abstract

The genus *Cetraria* s. str. (Parmeliaceae family, Cetrarioid clade) consists of 15 species of mostly erect brown or greenish yellow fruticose or subfoliose thallus. These *Cetraria* species have a cosmopolitan distribution, being primarily located in the Northern Hemisphere, in North America and in the Eurasia area. Phytochemical analysis has demonstrated the presence of dibenzofuran derivatives (usnic acid), depsidones (fumarprotocetraric and protocetraric acids) and fatty acids (lichesterinic and protolichesterinic acids). The species of *Cetraria*, and more particularly *Cetraria islandica*, has been widely employed in folk medicine for the treatment of digestive and respiratory diseases as decoctions, tinctures, aqueous extract, and infusions. Moreover, *Cetraria islandica* has had an important nutritional and cosmetic value. These traditional uses have been validated in in vitro and in vivo pharmacological studies. Additionally, new therapeutic activities are being investigated, such as antioxidant, immunomodulatory, cytotoxic, genotoxic and antigenotoxic. Among all *Cetraria* species, the most investigated by far has been *Cetraria islandica*, followed by *Cetraria pinastri* and *Cetraria aculeata*. The aim of the current review is to update all the knowledge about the genus *Cetraria* covering aspects that include taxonomy and phylogeny, morphology and distribution, ecological and environmental interest, phytochemistry, traditional uses and pharmacological properties.

## 1. Introduction

Lichens are the result of a symbiotic association between a fungus (mycobiont organism) and photoautotrophic organisms—algae and cyanobacteria (photobiont) [1]. These organisms result from a mutualistic relationship in which the photobionts provide its mycobiont with sugars and organic nitrogen for photosynthesis, and the fungus provides protection to the photobiont from very harsh weather conditions and mechanical damage [2]. In addition, in the last decade, lichen symbiosis has included the presence of bacterial associates (i.e., alphaproteobacterial communities in *Cetraria aculeata* and Acetobacteriae and Acidobacteriaceae communities in *Cetraria islandica*) [3,4].

It has been described that there are about 20,000 lichen species worldwide, with the Parmeliaceae family being the most numerous with more than 2700 species distributed in about 70 genera [1,5,6,7]. Within this diversity of genera, in this work, we will focus on *Cetraria* s. str. in its strict sense.

Etymologically, the term *Cetraria* comes from a diminutive of the Latin word *cetra*, a small and light leather shield of Roman fiction, and it refers to the morphological aspect of the papillae of these lichens [8]. The genus *Cetraria* s. str., belonging to the Cetrarioid clade, consists of 15 lichen species [9]. *Cetraria islandica* (L.) Ach. has been the most investigated species from the pharmacological perspective within this genus of *Cetraria* (Figure 1).

The species of the genus *Cetraria* s. str. has aroused great ecological, environmental, and therapeutical interest [9]. Thus, these species are good indicators of contamination via detecting, for example, high levels of sulfur dioxide and fluoride [10]. Moreover, these lichens also absorb radioactive fallout [10,11].

Regarding its therapeutic activity, the species of the genus *Cetraria* s. str., and more particularly, *Cetraria islandica* (L.) Ach., have been traditionally used in folk medicine to alleviate the symptoms of a variety of digestive and respiratory pathologies, such as cough, cold and constipation [12]. Moreover, consistent evidence has demonstrated various biological activities for *Cetraria* spp. extracts and its isolated secondary metabolites, such as antimicrobial, antioxidant, anticancer and antidiabetic [13,14,15,16]. Some of these activities are correlated with its folk uses, but there are other activities of drug repurposing that have roused a renewed and growing interest in lichens as a promising source of novel therapeutic actions. Furthermore, the existence of several patents on the therapeutic activities of *Cetraria islandica* is also worth mentioning. *Cetraria islandica* has been approved as a patent in veterinary medicine (i.e., ear hygiene), as a feedstuff for the respiratory health of horses (Patent number DE3229086 and US20030068294), as a wood protection product (Patent number WO2008077997), and to prevent and treat asthma (Patent number FR2756182) [17,18,19,20]. These beneficial effects are mainly attributed to the major secondary metabolites identified in lichens of the *Cetraria* genus, such as lichesterinic acid, protolichesterinic acid, fumarprotocetraric acid and protocetraric acid, among others [21].

The aim of the current review is to update all the knowledge about the genus *Cetraria* s. str. as a source of new drugs, covering aspects that include taxonomy and phylogeny, morphology and distribution, ecological and environmental interest, phytochemistry, traditional uses and pharmacological properties.

The data collected for this review were taken from electronic databases, including Pubmed, Google Scholar, Wiley, ScienceDirect, Springer and Cochrane. The keywords used were *Cetraria*, secondary metabolites, distribution, traditional uses, folk medicine, ecological, phytochemistry, chemical composition, pharmacology, and phylogeny. We selected all papers written in English, French and Spanish. No time restriction was set.

## 2. General Features

### 2.1. Taxonomy/Phylogeny

Due to the complexity of living organisms, and to understand and interpret their diversity, biological science relies on taxonomy, providing an organizational framework for other sciences and disciplines [22]. Since Linnaeus established the two fundamental rules of classification (binomial names and higher rank-based classifications) more than 250 years ago, some changes have been introduced in phylogenetic studies. To morphology, anatomy, biochemistry, and physiology criteria were added the techniques of molecular phylogenetics, which have changed the way researchers understand evolutionary relationships. Surprising findings have been made, especially in the fungi kingdom [23]. There have been selected specific nucleotide sequences of mycobiont ribosomal genes, such as the small subunit (SSU) and the large subunit (LSU) of the ribosomal DNA (rDNA) for higher taxonomic levels (family and above), and internal transcribed spacer (ITS) regions, PKS regions and introns (group I intron) for species and genera phylogeny studies [21,24]. Taxonomic ranks in a hierarchical classification are arbitrary, and lead to potential biases [25]. Hence, Hennig introduced the idea of the temporal banding approach because “time creates the systematic categories, not the arbitrariness of man” [26]. To increase taxonomic rank consistency, this approach gives the same rank if organisms share their origin in a standardized geological time [25]. However, some discrepancies have been found regarding the abuse of strict temporal banding, in which cases reviews improve existing classifications [23].

In this context, the *Cetraria* genus appears to be a discording genus in terms of classifications. *Cetraria* is one of the 70 genera belonging to the largest family of lichen-forming fungi, the Parmeliaceae. This family is divided into seven main clades: parmelioid, usneoid, anzioid, alectorioid, psiloparmelioid, hypogymnioid and cetrarioid, this very last one being where species of *Cetraria* s. str. are placed among another thirteen genera [9,27]. The genus was originally described by Acharius in 1803 [28]. Firstly, it was included in the Usneaceae family by Eschweiler (1824) [29]. *Cetraria* was a species-rich genus until the 1960s, when new genera were described (i.e., *Asahinea*, *Platismatia*) followed, by new descriptions in successive years (*Masonhalea*, *Ahtiana*, *Allocetraria*. *Vulpicida*, *Cetrariella*, *Arctocetraria*, *Flavocetraria*, among others). In the late 1990s, thanks to phylogenetics, the genus was reevaluated [9]. In the last few years, due to the temporal banding approach previously mentioned, Divakar et al. proposed to synonymize *Allocetraria*, *Cetrariella*, *Usnocetraria* and *Vulpicida* with *Cetraria*, with a temporal band between 29.45 and 32.55 Ma at the genus level [6]. As this latest classification has not yet been universally accepted (see, e.g., [23]), we have focused this review only on *Cetraria* s. str. species.

### 2.2. Morphology and Distribution

The first description of the *Cetraria* genus dates to 1803, when the Swedish botanist known as the father of lichenology, Erik Acharius, published his work Methodus qua omnes detectos lichenes ad genera redigere tentavit. In the first description, the lichen species of *Cetraria* genus were characterized by a foliose, membranous, cartilaginous thallus with irregular and curled lobes [28]. Over the following centuries, the morphological criteria have been modified with the inclusion and exclusion of species within this genus. Nowadays, the taxonomical discrepancies among the scientific community have an impact on the main morphological features used to identify the genus. The basic morphological characteristics used for the identification of *Cetraria* species include the color and growth form of the thallus, the presence of structures such as pycnidia, isidia, soredia and cilia, and the presence of pseudocyphellae on the upper or lower surfaces, or both. Reproductive structures such as apothecia and conidia are also essential for identification, with great variety among the different genus of the cetrarioid clade [30].

Species of *Cetraria* s. str. are characterized by erect fruticose or subfoliose thallus, dorsiventral with canaliculate lobes, brown and yellow pigmentation, and a lack of rhizines. The epicortex is nonpored and the punctiform to sublinear pseudocyphallae, when are present, are found marginal or laminal on the lower surface, not on the upper one. The reproductive organs are distinguished on the upper cortex by predominantly marginal and laminal apothecia, asci clavate, which usually produce eight subspherical ascospores (6–10 × 3–5 µm in size), terminal pycnidia on projections, and oblong, citriform, sublageniform or filiform conidia [6,31,32]. Within these general features, diverse species present their own particularities (Table 1).

There are different forms of thallus in the *Cetraria* species, including a fructicose dark brown thallus, such as in *Cetraria aculeata*, *Cetraria australiensis*, *Cetraria crespoae*, *Cetraria ericetorum*, *Cetraria muricata* and *Cetraria steppae*, among others; or a foliose yellow or greenish yellow thallus, such as in *Cetraria annae*. Moreover, marginal pseudocyphellae were abundant on the lower surface of some species, such as *Cetraria aculeata* or *Cetraria crespoae*; however, these structures were absent in other lichen species, such as *Cetraria sepincola*, or badly visible in *Cetraria kamczatica*, *Cetraria muricata* and *Cetraria nigricans*. The great variety between species is also seen through the presence/absence of reproductive structures, from species with marginal apothecia (*Cetraria nigricans*) to species with absent (*Cetraria peruviana*) or unknown ones (*Cetraria nepalensis*). Differences in soredia, isidia, conidia or pycnidia are also visible in Table 1 [30,33,34,35,36,37,38,39].

According to their habitat ecology and substratum, lichens can be classified as corticolous (on trees), ramicolous (on twigs), saxicolous (on rocks), lignicolous (on dead wood), terricolous (over soil), folicolous (on leaves) and muscicolous (on mosses) [40]. A large majority of *Cetraria* species are corticolous or terricolous, but some species, such as *Cetraria odontella* or *Cetraria ericetorum*, are saxicolous. They can also be muscicolous, which is the case with *Cetraria kamczatica* [33,34,37].

*Cetraria* s. str. is a cosmopolitan genus, occurring on all continents and many islands (Table 2). The largest number of species are predominantly distributed in the Northern Hemisphere, in North America and in the Eurasia area. The most widespread species are *Cetraria aculeata* (Schreb.) Fr. and *Cetraria muricata* (Ach.) Eckfeldt [41]. Some species follow more restricted patterns of distribution: bipolar, circumboreal, circumpolar, and amphi-Beringian. *Cetraria sepincola* and *Cetraria odontella* occur in boreal forests and tundras of the artic [41,42]. Amphi-beringian species, such as *Cetraria kamczatica*, *Cetraria laevigata* and *Cetraria minuscula*, are found in eastern Siberia and Alaska [36,37,43,44]. *Cetraria nigricans* is found in high arctic areas and alpine sites in southern areas, showing a circumpolar pattern [41,44]. Nevertheless, we found a great number of species restricted to certain regions that have been scarcely studied (i.e., *Cetraria crespoae* in western parts of the Iberian and Italian Peninsula, *Cetraria peruviana* in central parts of South America, *Cetraria australiensis* in South and Eastern Australia, and *Cetraria obstusata* in alpine areas of the Alps). Endemic *Cetraria* lichens are also found in some regions in Russia, such as the Baikal region (*Cetraria annae*, *Cetraria rassadinae*), or in semiarid steppe biomes, such as as Kazakhstan to Iran and Ukraine (*Cetraria steppae*) [30,38,45]. It is worth mentioning the species *Cetraria ericetorum*, whose subspecies are separated geographically, occurring in diverse areas around the globe (subsp. *ericetorum* in Eurasia, subsp. *reticulata* in North America and subsp. *patagonica* in the southern part of South America) [36,45].
molecules-27-04990-t001_Table 1Table 1Morphological features of species from the genus *Cetraria* s. str.
MorphologyType of ThallusColorStructuresSubstrateReferences*Cetraria aculeata* (Schreb.) Fr.FruticoseThallus: brown to black.Pseudocyphellae abundant Isidia and soredia absentTerricolous[35]*Cetraria annae* OxnerFolioseThallus: pale yellowMedulla: whiteSoredia are white, granularNo apothecia were seenMarginal black pycnidia Conidia absentTerricolous[30]*Cetraria australiensis* W.A.Weber ex KärnefeltFruticoseThallus: Upper surface dark. Lower surface yellowish brown to dark brown.Marginal pseudocyphellaeMarginal projections with terminal pynidiaApothecia not knownTerricolousCorticolous[30]*Cetraria crespoae* (Barreno and Vázquez) KärnefeltFruticoseThallus: olive or brown (to almost black).Soredia absentAbundantPseudocyphellaeCorticolous[39]*Cetraria ericetorum* OpizFruticoseThallus: dark brown to paler brownMedulla: whitePseudocyphellae abundant on the marginsApothecia frecuentSoredia absentPycnidia laminal/marginalTerricolousCorticolousSaxicolous[39]*Cetraria islandica* (L.) Ach.FolioseThallus: upper surface greenish or greenish-brown, lower surface is greyish-white or light brownish.Laminal pseudocyphellaepresent but sometimes badly visibleRarely discoid apothecia on terminal lobesPycnidia presentTerricolous[12,41]*Cetraria kamczatica* SaviczFruticoseThallus: dark brown.Medulla: white.Apothecia not seenNo pseudocyphellae or with few tiny pseudocyphellae on lower marginsMusciolous[37]*Cetraria laevigata* Rass.FruticoseThallus: upper side pale brown, underside paler.Apothecia not seenMarginal pseudocyphellaeTerricolous[37]*Cetraria muricata* (Ach.) EckfeldtFruticoseThallus: Brown toblack.Pseudocyphellae scattered,poorly visible, depressedIsidia and soredia absentTerricolous[41]*Cetraria nepalensis* D.D. AwasthiFructicoseThallus: brown to black.Apothecia unknownTerricolous[45]*Cetraria nigricans* Nyl.FolioseThallus: upper side dark brown or olive-brown, lower side pale brown.Apothecia presentPseudocyphellae absent or badly visiblePycnidial projectionsMarginal cilia numerous and longTerricolousSaxicolous[41,44]*Cetraria odontella* (Ach.) Ach.FruticoseThallus: olive or brown (to almost black).Pseudocyphellae exclusively on the undersideEpilithic[34]*Cetraria obtusata* (Schaer.) Van den Boom and SipmanFruticoseThallus: Dark brown.Medulla: Pale yellow.Pseudocyphellae presentApothecia unknownPycnidia darkConidia clavateTerricolous[34,38,44]*Cetraria peruviana* Kärnefelt and ThellFruticoseThallus: reddish, brown to dark brown or almost black.Apothecia usually absentMarginal ciliaPycnidial projections absentTerricolousSaxicolous[41]*Cetraria rassadinae* MakryiFruticoseThallus: brownish blackPycnidium presentConidia oblongTerricolous[36]*Cetraria sepincola* (Hoffm.) Ach.FruticoseThallus: brown to almost blackMarginal pseudocyphellae absent.Absent soredia and isidiaCorticolousLignicolous[34]*Cetraria steppae* (Savicz) KärnefeltFruticoseThallus: black, brown to light brownMedulla: whitePseudocyphellae depressed or poorly visibleIsidioid projections absentTerricolous[34,35]


The distribution patterns of *Cetraria aculeata* have been thoroughly investigated. Using phylogenetic methods, the bipolar pattern of *C. aculeata* appears to have its origin in the Northern Hemisphere and was later dispersed to Antarctica and South America during the Pleistocene [46]. Phenotypic variations have also been identified in *C. aculeata* species from the steppe regions of Spain [47].

Among all the described species, *Cetraria islandica* occupies the most outstanding place within the genus and is one of the most important at the pharmacological level within the lichens. *Cetraria islandica* is characterized by a foliose, dichotomous thallus, a length up to 15 cm, and a greenish or greenish brown upper surface and greyish white or brownish lower surface, with laminal pseudocyphellae and marginal pycnidia. Brown discoid apothecia are rarely found on terminal lobes. This terricolous lichen is distributed in high latitudes in the Northern and Southern Hemispheres (subsp. *islandica*), in northern and eastern Europe, Siberia and North America (subsp. *crispiformis*) and in the Southern hemisphere (subsp. *antartica*) [12,36,41].

## 3. Ecological and Environmental Interest of *Cetraria* s. str. Species

To assess the biological impact of air pollutants, not only can monitoring be done using instrumental devices, but biomonitoring is also essential, especially in the case of the long-lasting effects of pollution events [48].

The air-dependent nutrition of lichen species and the lack of cuticles in higher plants increases their sensitivity to atmospheric changes and their absorbance of air pollutants, making them great indicators of pollution and climate change. Historically, the use of lichen for environmental biomonitoring has been widely adopted because it is inexpensive and applicable on a large scale [49]. Climate change monitoring, on the other hand, has its difficulties. To exclusively monitor the effect of climate change on lichens, study areas must be free from other factors, such as pollution or increased collection for commercial interest [50].

Some current studies are extending the uses of lichens as biomonitors in indoors environments (i.e., a shooting range) for assessing the accumulation of heavy metals, including cadmium, copper, iron, manganese and nickel, among others, and therefore for studying health consequences [51,52].

There are different approaches to biomonitoring with lichens: analyzing trace element bioaccumulation, studying the biodiversity of lichens in certain areas, and focusing on the integrity of cell membrane [53,54].

To date, several *Cetraria* species have been used as bioindicators of air pollutant deposition. *Cetraria islandica* has been proven to be a bioaccumulator of elements such as Al, Cr, Li, Mg, Cd, Hg and Pb, which are also known as potentially toxic elements (PTEs) [49]. Non-living *Cetraria islandica* have also shown the capacity for the biosorption of Au (III) and Cu (II), with values of 7.4 mg/1 g dried lichen and 19.2 mg/1 g, respectively [55]. Contrary to this, depending on the pH of the environment, lichens are able to desorb some metals in acid rain simulated conditions [56].

Elements’ occurrence in lichens is influenced by natural processes, such as soil erosion (mainly Al, Cr, Li and Mg) and anthropogenic origins (Cd, Hg or Pb). Some have found diverse patterns by analyzing elemental deposition. In Mediterranean ecosystems, specifically in eastern areas of Majella massif (an unpolluted area at 2793 m), high-elevation areas showed increased levels of PTEs, which can be explained by processes such as long-distance transport and accumulation by cold condensation. Hence, the differences among eastern and western areas have allowed us to suggest the transport of air pollutants from Balkan areas [49]. In Indian Himalayan areas, 245 macrolichen species, including *Cetraria nigricans* Nyl., were used to evaluate pollution and anthropogenic disturbances [57]. The lichen diversity in the Badrinath site showed less pollution and less human impact [58]. Moreover, the analysis of trace element concentrations in *Cetraria islandica* samples of Italian herbaria collected over more than 25 years revealed the presence of metals in lichens, especially those derived from human settlement (i.e., the variety in percentages of Pb in lichens collected next to roads related with Pb concentration gasoline reduction over the years) [59].

Furthermore, *Cetraria islandica* served as a bioindicator of contamination after the Chernobyl disaster by measuring the levels of 137Cs activity [60]. Moreover, Marchart et al. also studied the ecological half-lives of 137Cs in *Cetraria islandica* and *Cetraria aculeate*, with average values of 3 and 4 years, respectively [61]. Nevertheless, *Cetraria islandica* accumulate elements in lower proportions than *Evernia prunasti* and *Ramalina farinacea*, these last lichen species being more suitable for biomonitoring [62].

This useful capacity of lichens for bioaccumulation must be studied from a health perspective. As we have stated before, there are several patents and herbal preparations in the market containing *Cetraria islandica* lichen due to its medicinal properties [12,18]. The risk of contamination by trace element must be assessed in these preparations before they are placed on the market. In fact, *Cetraria islandica* is included in the Compendium of botanicals of possible concern for human health when used in food and food supplements, established by the European Food Safety Authority [63]. The European Community also established maximum concentration limits that cannot be surpassed in food preparations [64]. Some have identified trace elements in previously comminuted and decocted samples, and in dried samples of *Cetraria islandica*, with higher values in those samples from herbalist shops than those from companies, determined by using atomic emission spectrometry (ICP-OES). Nevertheless, decoction samples showed a significant reduction in elemental concentration, reaching favorable health levels in all the samples [65]. In another study, element and radiological levels have been characterized in *Cetraria islandica* food supplementation products using Energy Dispersive Polarized X-ray Fluorescence Spectrometry and alpha spectrometry; although the values were below tolerable doses, it is recommended to monitor the contaminants in these products for the sake of health assurance [66].

The role of *Cetraria* species in monitoring climate was also observed. Lichen communities can suffer alterations due to climate change. The ecophysiological processes in lichens are affected by temperature changes, UV exposure, ozone and humidity, resulting in lichen diversity patterns [67]. For instance, after analyzing 100 species in north-western Germany over the years, acidophilic lichens reduced their proportion, while nitrophytes species did not suffer exceptional changes. Moreover, some lichens that have never growth in these areas before seem to be increasing due to climate changes (i.e., higher temperatures) [68]. In Western Europe, while the numbers terricolous species seem to be declining because of climate change, the epiphytic species are increasing [69]. In terms of genetic structure, *Cetraria aculeata* species are associated with different photobiont lineages depending on geographical and ecological factors [70]. There is a concern about the survival of Antarctic populations of mycobionts and photobionts of *Cetraria aculeata* due to global warming, based on their low genetic diversity [71].

## 4. Traditional Uses

Among the species of the genus *Cetraria*, *C. islandica* (L.) Ach. has been the most widely used in traditional medicine. Its folk uses have been reported in a variety of documents, including handbooks, pharmacopoeias, compendia and pharmacognostical texts [12]. The main uses for *C. islandica*, commonly known as Iceland moss, are for the treatment of digestive and respiratory diseases. Herbal preparation varies from decoctions, tinctures and aqueous extracts to infusions. Hence, in Iceland, *C. islandica* has been used to the relief of both gastric and duodenal ulcers [72]. Moreover, decoctions of *C. islandica* were used to treat colds in Finland [73]. Furthermore, for centuries, *C. islandica* was famous as a laxative and antitussive in Central Europe. In addition to uses to treat respiratory and digestive conditions, *C. islandica* has been used in other countries for other medical purposes. This is the case in Sweden, where it has been used to treat nephritis and diabetes [74], and in Turkey, where it has been employed as a hemostatic and antihemorrhoidal agent [75,76]. Moreover, *C. islandica* has been used for tuberculosis in several countries including Spain, France, and Turkey [73,76,77]. These medicinal properties have been attributed mainly to its lichen acids, such as fumaroprotocetraric acid (2.6–11.5%), protocetraric acid (0.2–0.3%), protolichesterinic acid (0.1–1.5%) and usnic acid (0.04%) [78,79].

Based on the therapeutic benefits of *C. islandica*, comminuted herbal substances and soft extracts of this lichen in the form of syrup, oral gum, and lozenges have been marketed to relieve dry and irritating coughs and hoarseness. Moreover, combined commercial drug products have been developed that contain *C. islandica* together with other medicinal plants (*Thymus vulgaris* L., *Hyssopus officinalis* L., *Saponaria officinalis* L. and *Marrubium vulgare* L., among others) for the inflammatory processes of the upper airway, for the management of bronchial secretions and for the alleviation of coughs [12].

Other ethnobotanical studies have revealed traditional uses for other *Cetraria* species. Hence, in the Catalan Pyrenees (Spain), *Cetraria cucullata* (Bellardi) Ach. has been employed for asthma [77].

Many of the traditional uses of *Cetraria* spp. have been validated, such as antidiabetic and anti-inflammatory, and other new pharmacological activities are being investigated, such as cytotoxic and genotoxic/antigenotoxic, which we will discuss later in the section of pharmacological activities.

In addition to the medicinal uses of *C. islandica*, its nutritional value is noteworthy. It is mainly consumed as tea. Moreover, in Italy, it is a food supplementation product valued for its digestive-facilitating properties [12]. In Northern Europe, during times of famine, it was used for bread, sometimes mixed with rice or flour [80,81]. Furthermore, *C. islandica* is approved as a flavoring for alcoholic beverages in United States [82]. In Iceland, *C. islandica* is used in a variety of recipes including soups, porridges and sausages, and is added to “skyr” (curd) [79], and it is also marketed as a bitter alcoholic beverage (38% alcohol content) called “*Cetraria islandica* schnapps” [81]. On the other hand, in Russia, during the years 1942–1943, *C. islandica* was used to industrially extract glucose because of the beet sugar scarcity [81]. Moreover, the lichen species *Cetraria ericetorum* Opiz was chopped up and added to soups for flavoring [83]. In addition to food uses in humans, the species *C. islandica* has been used as food for pigs and cows, especially during World War II [83]. Finally, it is also worth mentioning its uses in cosmetics. *C. islandica* is part of the composition of diversity cosmetic products, including shampoos and conditioners, deodorants, toothpastes, exfoliating and anti-cellulite creams, manicure and pedicure products, and aftershave lotions. Moreover, the lichen *Cetraria nivalis* (L.) Ach. is used in rejuvenating cream formulations [84].

## 5. Phytochemistry

The development of experimental and analytical techniques has enabled the identification and characterization of lichen compounds. Chemical profiling using new techniques, such as Ultra-high Performance Liquid Chromatography with Quadrupole Time-of-Flight Mass Spectrometry (UPLC-QToF-MS), combined with DNA barcoding, has been a useful approach for lichen authentication. For example, these techniques allow us to distinguish between the two *Cetraria islandica* chemotypes, and both from *C*. *ericetorum* [79].

Based on their biochemistry, lichens produce primary metabolites, which are mainly nonspecific from lichens and essential for lichen physiology (i.e., proteins, aminoacids, polyols, and polysaccharides) and secondary metabolites. Secondary metabolites are mainly composed of organic polycyclic compounds synthetized through one of these three pathways: the mevalonic acid pathway, the shikimic acid pathway and the acetyl-polymalonyl pathway [85]. More than 1000 lichen substances have been identified [86]. These substances are scarcely found in higher plants or other fungi, being mostly unique and specific to the lichens [87]. Historically, these substances were known to be produced by the fungal partner, but in recent years phytochemical studies of the photobionts and lichen-associated bacteria have evidenced that these partners also play an important role and produce substances that benefit the lichen [88].

Within the secondary metabolites from the *Cetraria* species, the most abundant ones are polyketides and aliphatic acids. Dibenzofuran derivatives such as usnic acid, depsidones such as fumarprotocetraric and protocetraric acids, and fatty acids such as lichesterinic and protolichesterinic acids, are presented in *Cetraria* species (Figure 2, Table 3). Variations are found among species, and even between subspecies. This is the case for the fumarprotocetraric acid content, which is found in higher concentrations in *Cetraria islandica* ssp. *islandica* than in ssp. *crispiformis* [89]. Among these metabolites, usnic, fumarprotocetraric, protocetraric, and protolichesterinic acids stand out for their potential biological applications [13,90,91,92].

Conversely to the phytochemistry of other lichen genus as *Parmotrema* or *Usnea*, the *Cetraria* genus show less variations in their interspecies composition [93,94]. The chemical composition of the *Parmotrema* species showed the presence of didepsides such as atranorin in *P. perlatum* (Huds.) M. Choisy, lecanoric acid in *P. andinum* (Müll. Arg) Hale, and alectoronic acid and atranorin in *P. nilgherrense* (Nyl.) Hale [93]. However, in *Cetraria* species, depsides such as atranorin and squamatic acid were only found as minor compounds in *Cetraria annae* Oxner [30], and the compound norstictic acid in *Cetraria steppae* (Savicz). In the case of norstictic acid, its presence in *Cetraria steppae* was crucial to delimitating this endemic species from cosmopolitan lichen *Cetraria aculeata* (Schreb.) Fr. Using HPLC techniques, traces of norstictic acid were detected also in *Cetraria aculeata*. Environmental factors such as climate and soils determine variations in concentration among individuals. Norstictic acid was only presented in the *Cetraria aculeata/steppae* (not in *C*. *muricata*) species that grow in the Mediterranean and Central Asian [35,95].

In addition to protolichesterinic and lichesterinic acids, other less common fatty acids have been identified. These non-polyketides are abundant in this genus. As examples, *Cetraria nigricans* Nyl. and *Cetraria odontella* (Ach.) Ach were synthesized from rangiformic acids. Secalonic acid has been identified in *Cetraria obtusata* (Schaer.) Van den Boom and Sipman [34,44].

Among all *Cetraria* lichen species, the depth of knowledge about *Cetraria islandica* surpasses every other species of the genera. Not only are there data about its secondary metabolites (Table 3), but its primary metabolites are also characterized. Lichenan (β-1,3/1,4-D-glucan) and isolichenan (α-1,3/1,4-Glucan) stand out as the main glucan components of *C. islandica* whose yield of extraction depends on pH and temperature [96,97]. Moreover, the ratio of β-1,3/1,4-D-glucans is higher in the mycobiont cell wall of *Cetraria islandica* than in barley and oats [98]. Other polysaccharides have already been isolated from this lichen species, such as alkali-soluble galactomannan, and several soluble polysaccharides [99,100,101].

The symbiotic association, found in lichens, is a type of coevolution that allows these species to be distributed in a cosmopolitan pattern from polar to desert areas. Recent studies suggest that the production of secondary metabolites plays an important role in their environmental adaptation, and thus in their global distribution [102]. For instance, fumarprotocetraric acid from *Cetraria islandica* is related with the heavy metal tolerance of lichens due to its role in their reduction of metal ion absorption in the apoplast. Moreover, fumarprotocetraric acid plays a role in SO_2_ pollution tolerance [103,104]. Furthermore, novel studies have focused on lichens’ dark pigments, melanins (i.e., allomelanins isolated from *C. islandica*), which are essential to UV protection. Rassabina et al. (2020) suggested that these melanins could influence lichens’ survival at harsh conditions [105]. In 2004, Nybakken et al.’s study introduced the importance of melanin as a UV screening pigment in *Cetraria islandica*, showing that it absorbs both UV-B and photosynthetically active radiation. In this work, although the French Alps species of *C. islandica* that were exposed to high ambient UV-B showed higher transmittance, all samples, even in low UV-B ambient, were protected [106].

## 6. Therapeutic Potential

The pharmacological activity of *Cetraria* species is summarized in Table 4.

### 6.1. Antibacterial, Antifungal and Antitrypanosomal Activities

The antibacterial and antifungal activities of species of the genus *Cetraria* have been investigated for *Cetraria pinastri*, *Cetraria aculeata* and *Cetraria islandica*. These lichen species have shown antibacterial activity against a broad group of both Gram-positive and Gram-negative bacteria. Hence, the Gram-positive microorganisms sensitive to the methanol extract of *Cetraria pinastri* were *Enterococcus fecalis* (Minimum Inhibitory Concentration (MIC) 0.23 mg/mL), *Micrococcus lysodeikticus* (MIC 0.46 mg/mL) and *Staphylococcus aereus* (MIC 0.94 mg/mL). The only Gram-negative bacteria that was sensitive to the methanol extract of *Cetraria pinastri* was *Escherichia coli*, with an MIC value of 1.87 mg/mL [107]. In another study, the antimicrobial activity of the ethanol, acetone and diethylether extracts of *Cetraria acuelata* against 12 different Gram-positive (*Bacillus cereus*, *Bacillus subtilis*, *Listeria monocytogenes*, *Staphylococcus aureus and Streptococcus faecalis*) and Gram-negative (*Escherichia coli*, *Klebsiella pneumoniae*, *Pseudomonas aeruginosa*, *Pseudomonas syringae*, *Proteus vulgaris*, *Aeromonas hydrophila* and *Yersinia enterocolitica*) bacteria was investigated. The three extracts tested showed antimicrobial activity against all bacteria species, except for the Gram-negative bacteria *Pseudomonas syringae*, *Klebsiella pneumoniae* and *Yersinia enterocolitica*. The most potent of the three tested *Cetraria aculeata* extracts was the diethylether extract, being especially active against the Gram-positive bacteria *Bacillus cereus*, *Bacillus subtilis*, *Listeria monocytogenes* and *Streptococcus faecalis*, and against the Gram-negative bacteria *Pseudomonas aeruginosa* and *Proteus vulgaris*, with MIC values of 8460 µg/mL [13].

Regarding the antimicrobial activity of *Cetraria islandica*, the extracts of methanol, acetone, water, and light petrolatum were assayed against *Helicobacter pylori* by using the Kirby and Bauery disk diffusion test. The light petrolatum extract of *Cetraria islandica* was the one that showed the greatest inhibitory capacity, followed by the acetone extract. However, the methanol and water extracts showed no activity. In addition, it was demonstrated that the compound responsible for the antimicrobial activity in the light petrolatum extract was the protolichesterinic acid [108]. In another study, the antimicrobial activity of the methanol extract of *Cetraria islandica* against Gram-positive species such as *Staphylococcus aureus*, *Bacillus subtilis* and *Bacillus cereus*, and against Gram-negative species such as *Escherichia coli* and *Proteus mirabilis*, was evaluated. The results show that the methanol extract of *Cetraria islandica* was more active against *Escherichia coli* with an MIC value of 2.5 mg/mL, followed by *Staphylococcus aureus* (MIC 1.25 mg/mL), *Proteus mirabilis* (MIC 1.25 mg/mL), *Bacillus subtilis* (MIC 0.625 mg/mL) and *Bacillus cereus* (MIC 0.312 mg/mL) [15].

In addition to studies evaluating the antibacterial activity of the lichen extracts from *Cetraria* species, the antimicrobial action of the compound protolichesterinic acid, isolated from *Cetraria*, *Cetraria aculeata* and *Cetraria islandica*, was investigated against different types of Gram-positive and Gram-negative bacteria. Hence, in one study, the protolichesterinic acid was active against *Escherichia coli* (MIC 7341 µg/mL), *Bacillus subtilis* (MIC 7341 µg/mL), *Pseudomonas aeruginosa* (MIC 7341 µg/mL) and *Listeria monocytogenes* (MIC 3670 µg/mL) [13]. In another study, the inhibitory capacity of the protolichesterinic acid was evaluated against a total of 35 *Helicobacter pylori* strains randomly selected from human biopsy samples, obtaining MIC values ranging from 16 to 64 µg/mL [108].

Apart from their antibacterial activity, *Cetraria* species, specifically *Cetraria aculeata*, *Cetraria pinastri* and *Cetraria islandica*, have been found to be active against pathogenic fungi. Hence, the methanol extract of *Cetraria pinastri* showed antifungal activity against *Acremonium chrysogenum* (MIC 3.75 mg/mL), *Alternaria alternata* (MIC 1.87 mg/mL), *Aspergillus flavus* (MIC 7.5 mg/mL), *Aspergillus niger* (MIC3.75 mg/mL), *Candida albicans* (MIC 1.87 mg/mL), *Cladosporium cladosporioides* (MIC 0.94 mg/mL), *Fusarium oxysporum* (MIC 7.5 mg/mL), *Mucor mucedo* (MIC 7.5 mg/mL), *Paecilomyces variotii* (MIC 15 mg/mL), *Penicillium verrucosum* (MIC 15 mg/mL) and *Trichoderma harsianum* (MIC 3.75 mg/mL). In another study, the ethanol, acetone, and diethyl ether extracts of *Cetraria aculeata* were assayed against the fungal species *Penicillum* sp., *Cladosporium* sp., *Fusarium oxysporum*, *Fusarium culmorum*, *Rhizopus* sp., *Fusarium moniliforme*, *Fusarium solani* and *Aspergillus* sp. The results show that none of the extracts of *Cetraria aculeata* had antifungal activity [13,107]. Finally, the methanol extract of *Cetraria islandica* was investigated on *Aspergillus flavus*, *Candida albicans*, *Fusarium oxysporum*, *Penicillium purpurescens* and *Trichoderma harsianum* species. *Cetraria islandica* showed a high antifungal activity against *Aspergillus flavus* and *Penicillium purpurescens*, with MIC values of 5 mg/mL, followed by its activity against Fusarium *oxysporum* and *Trichoderma harsianum*, with MIC values of 2.5 mg/mL. The lowest antifungal activity of the methanol extract of *Cetraria islandica* was against *Candida albicans* (MIC 1.25 mg/mL) [15]. 

In addition to the antifungal and antibacterial activity studies, the antitrypanosomal activity against *Trypanosoma brucei brucei* of bioactive compounds isolated from *Cetraria islandica* species was also evaluated. Protolichesterinic acid, fumarprotocetraric acid, lichesterinic acid and protocetraric acid were isolated. It was observed that only protolichesterinic acid (MIC 12.5 µM) and lichesterinic acid (MIC 6.30 µM) showed antitrypanosomal activity, with protolichesterinic acid being more effective [109].

### 6.2. Antioxidant Activity

Oxidative stress is characterized by an imbalance between the production of reactive species (ROS) and the antioxidant defense activity. This pathological state has been associated with many chronic and/or degenerative diseases, such as diabetes, Alzheimer’s disease, and cardiovascular disease. The use of exogenous antioxidants that act as scavengers or that modulate the endogenous antioxidant system is one of the most promising therapies used to deal with oxidative stress. In this context, phenolic compounds have turned out to be powerful antioxidants. The efficacy is directly related to the number of hydroxyl groups in the phenolic structure [110]. Lichens produce unique phenolic compounds as secondary metabolites that have aroused great research interest due to their antioxidant capacity [111].

Ranković B et al. evaluated the antioxidant activity of the methanol extract of *Cetraria pinastri* species by measuring the oxidation products of linoleic acid. The results have revealed that this lichen species was able to inhibit linoleic acid oxidation by 48.8%. This activity is related to its high polyphenol content of 32.9 mg/g in the dry extract [107].

Likewise, antioxidant activity studies have been carried out with the methanol extract and the ethyl acetate extract of *Cetraria aculeata*. In general, the antioxidant capacity of the methanol extract was higher than that of the ethyl acetate extract. Hence, the methanol extract showed an IC_50_ value of 51.65 µg/mL in the 2,2-diphenyl-1-picrylhydrazyl (DPPH) assay, a value of 45.55 µg/mL for lipid peroxidation inhibition capacity, a value of 50.43 µg/mL for ferrous ion chelating capacity and a value of 90.1 µg/mL for hydroxyl radical scavenging activity. Regarding antioxidant enzymes, the methanol extract of *C*. *aculeata* increased superoxide dismutase (SOD) and glutathione peroxidase (GPx) enzymes’ activities. On the contrary, a slight increase in malondialdehyde (MDA) levels and a decrease in reduced glutathione (GSH) levels were also observed at low doses [112].

On the other hand, the lichen in which the antioxidant activity has been the most studied is *Cetraria islandica*. Hence, the methanol extract of *Cetraria islandica* exhibited a high reduction capability and powerful free radical scavenging, as shown in the DPPH assay (IC_50_ 678.38 μg/mL), superoxide anion assay (IC_50_ 792.48 μg/mL) and reducing power assay (from 0.0512 μg/mL to 0.4562 μg/mL) [15]. In another study, Kotan E. et al. investigated the antioxidant ability of the methanol extract of *Cetraria islandica* (5 and 10 µg/mL) in an aflatoxin B1 induced oxidative stress model in blood lymphocytes from healthy non-smoking volunteers. The methanol extracts of this lichen species increased SOD and glutathione peroxidase (GPx) enzyme activities, and decreased MDA levels. The most protective concentration of *Cetraria islandica* was 5 µg/mL [113]. In addition to the methanol extract, the aqueous extract of *Cetraria islandica* has shown antioxidant activity using the thiocyanate method and the reducing antioxidant power, superoxide anion radical and DPPH assays. Different concentrations of the aqueous extract of *Cetraria islandica* (50, 100 and 250 µg) inhibited the peroxidation of linoleic acid by 96 to 100%. Moreover, this extract reduced iron from its ferric state to its ferrous state in a significant and concentration-dependent manner, this activity being higher than in the reference compound BHT. Moreover, the aqueous extract of *Cetraria islandica* at a concentration of 100 μg showed a higher superoxide radical scavenging activity than the reference compounds hydroxybutylanisole (BHA), butylated hydroxytoluene (BHT), and quercetin [114]. Furthermore, the ethanol extracts of *Cetraria islandica* (96%, 70% and 40%) showed antioxidant activity. The DPPH assay showed that the ethanol extract 70% and ethanol extract 40% had IC_50_ values of 2.40 mg/mL and 2.45 mg/mL; the Ferric Reducing Antioxidant Power (FRAP) assay revealed that the ethanol extract 96% had the highest value of 486 μmol/L, and the 2, 2′-Azinobis-3-ethyl-benzo-thiazoline-6-sulphonic acid (ABTS) assay demonstrated that the ethanol extract 40% of this lichen was the most active [115]. In another study, Kosanic M. et al. compared the antioxidant activities of different extracts (methanol, water, and acetone) of *Cetraria islandica* by using the DPPH assay, reducing antioxidant power method and superoxide anion radical assay. This study revealed that the methanol extract had the highest activity in all these methods, followed by the acetone extract and aqueous extract [116]. Recently, it has been observed that melanin extracted from *Cetraria islandica* also possesses free radical scavenging and reducing capacity in the DPPH assay, with an IC_50_ of 405 μg/mL [105].

Finally, based on the antioxidant properties and considering the implications of oxidative processes in diabetes and neurodegenerative diseases, there are several studies focused on the protective effect of *Cetraria islandica*. Hence, the aqueous extract of this lichen species increased the activity of the antioxidant enzymes SOD, catalase (CAT), and GPx, and reduced the levels of the lipid peroxidation biomarker MDA in human erythrocytes with type 1 diabetes mellitus [117]. Moreover, the aqueous extract of *Cetraria islandica* decreased the total oxidative stress (TOS) and increased the total antioxidant capacity (TAC) in streptozotocin-induced Diabetes Mellitus type 1 Sprague-Dawley rats [118]. Using the same model, the aqueous extract of *Cetraria islandica* (250–500 mg/kg/day) increased the levels of SOD, CAT and GSH, and reduced the levels of MDA [16,119,120]. Regarding neuroprotection studies, the protective role of *Cetraria islandica* and its isolated secondary metabolite fumarprotocetraric acid has been demonstrated in a hydrogen peroxide-induced oxidative stress model on human U373MG astrocytoma cells and human SH-SY5Y neuroblastoma cells. The methanol extract of *Cetraria islandica* at 10 µg/mL concentration increased cell viability, reduced intracellular ROS production and MDA levels, and increased the ratio of reduced/oxidized glutathione (GSH/GSSG). The depsidone fumarprotocetraric acid at 1 μg/mL in the neurons model and 25 μg/mL in the astrocytes decreased lipid peroxidation levels and intracellular ROS production and increased the ratio GSH/GSSG. Moreover, this compound ameliorated the H_2_O_2_-induced mitochondrial dysfunction and alterations in calcium homeostasis and inhibited apoptotic cell death. Its neuroprotective activity is related, at least in part, to its ability to activate the Nrf2 pathway that regulates antioxidant enzymes. Furthermore, both the lichen extract and fumarprotocetraric acid showed scavenging activities in the oxygen radical absorbance capacity (ORAC) assay (value of 3.06 µmol TE/mg and 5. 07 μmol TE/mg, respectively) and in the DPPH assay (IC_50_ value of 1183.55 µg/mL and 1393.83 μg/mL, respectively) [121,122].

### 6.3. Immunomodulatory and Anti-Inflammatory Activities

The immunomodulatory activity has been investigated on *Cetraria islandica* aqueous extracts and the isolated compounds fumarprotocetraric acid and protolichesterinic acid, and the polysaccharides lichenan and isolichenan on the maturation of dendritic cells. This study showed that the aqueous extract and the polysaccharide lichenan reduced the IL-12p40/IL-10 ratio and CD209 expression and increased CD86 expression. Moreover, this lichen extract showed anti-inflammatory properties at a dose of 2.5 mg/kg on a BSA-induced arthritis rat model, as evidenced in the reduction in the diameter between the right and left knees [90].

It is also noteworthy that the study on the polysaccharide α-1,3/1,4-D-Glucan (Ci3) isolated from *Cetraria islandica* (100 ug/mL) showed an increase in granulocytic phagocytosis and a decrease in complement-induced hemolysis [123].

### 6.4. Cytotoxic, Genotoxic and Antigenotoxic Activities

The cytotoxic activities of the lichen species *Cetraria acuelata* and *Cetraria islandica* and their isolated bioactive compounds have been evaluated on different types of malignant cells. Hence, the methanol extract of *Cetraria islandica* reduced the cell viability of the human breast cancer cells (MCF-7) (IC_50_ value of 19.51 µg/mL), the human liver cancer cells (HepG2) (IC_50_ value of 181.05 µg/mL), human melanoma cells (FemX) (IC_50_ value of 22.68 μg/mL) and human colon carcinoma cells (LS174) (IC_50_ value of 33.74 μg/mL) [15,121]. On the other hand, the ethanol extract of *Cetraria islandica* was able to reduce cell viability in MCF-7 cells (IC_50_ 9.2047 × 10^−5^ g/mL), also showing an increase in protein levels of AMP-activated kinases-α1 (AMPK-α1) and ERK1/2 [124]. The compound protolichesterinic acid, isolated from *Cetraria islandica*, decreased cell viability and caused morphological changes at a concentration of 20 µg/mL on breast carcinomas T-47D and ZR-75-1 and erythro-leukaemia K-562 cells. Moreover, this secondary metabolite inhibited DNA synthesis at the concentrations of 1.1 µg/mL on ZR-75-1 cells, 3.8 µg/mL on T-47D cells and 11.2 µg/mL on K-562 cells. This activity is related to the ability of protolichesterinic acid to inhibit 5-lipoxygenase [125]. Thorsteinsdottir et al. conducted a study in which it was observed that protolichesterinic acid, in addition to reducing human lung cancer cells’ (A549) viability, also induced a decrease in Leucine Rich Repeat Containing 8 VRAC Subunit A (LRRC8A) protein expression, as well as volume-sensitive taurine release under hypotonic conditions [126]. On the other hand, the fumarprotocetraric acid, isolated from *Cetraria islandica*, did not inhibit the cell growth of the human cells T-47D (breast) and Panc-1 (pancreas) [127]. Moreover, the lichenan from *Cetraria islandica* was also not active against the human myeloid leukemia U937 cells [128].

In addition to *Cetraria islandica*, the cytotoxic activity of *Cetraria aculeata* has been studied. The acetone extract of this lichen species was active against HeLa (human uterus carcinoma) (IC_50_ value of 200 μg/mL), A549 (human small lung carcinoma) (IC_50_ value of 500 μg/mL) and 5RP7 (c-H-ras transformed-rat embryonic fibroblasts) (IC_50_ value of 280 μg/mL) [129].

Regarding the studies performed to evaluate the genotoxic/antigenotoxic activities of *Cetraria* species, it has been highlighted that the species *Cetraria aculeata* showed a significant antigenotoxic effect on TA98 and TA100 strains of *Salmonella typhimurium but* had no effect in human lymphocytes [129]. On the other hand, the methanol extract showed antigenotoxic activity against *Salmonella typhimurium* strains TA1535 and TA1537, and a slight decrease in sister chromatid exchange (SCE) formation [14]. In another work, it has been shown that the methanol extract of *Cetraria islandica* (from 50 to 200 μg/mL) had genotoxic potential in cultured peripheral venous blood from healthy donors by increasing both the number of BN cells containing micronuclei (MNi) and the number of MNi in BN cells [15].

### 6.5. Cell Differentiation and Depigmentation Activities

The effect of β-1,3/1,4-Glucan (Lichenan), isolated from *Cetraria islandica*, on cell differentiation has recently been evaluated in primary normal human epidermal keratinocytes (NHEK) and HaCaT keratinocytes using immunofluorescence. A decrease in cell proliferation and an increase in protein expression of specific differentiation-related markers such as cytokeratin 10 (CK10) and involucrin were observed in NHEK cells. In addition, the gene expressions of CK, involucrin, transglutaminase, loricrin and filaggrin genes, which are involved in cell differentiation, were also increased [97].

The depigmenting activity was evaluated in chloroform, chloroform–methanol, methanol, and water extracts of *Cetraria islandica* species. The chloroform–methanol extract showed the highest inhibition capacity on tyrosinase (IC_50_ 86 µg/mL). A cell viability assay was performed on human melanoma cells (MeWo) with IC_50_ values of 264 µg/mL. Melanin assays demonstrated a significant reduction in melanin levels. On the other hand, zebrafish embryo models were used to determine the melanogenesis effects of the extract showing an inhibition of melanogenesis, and therefore a reduction in pigmentation [130].
molecules-27-04990-t004_Table 4Table 4Pharmacological activities of *Cetraria* spp.Lichen SpeciesExtracts/Active CompoundsExperimental ModelActivitiesResultsReferences*Cetraria aculeata*(Schreb.) Fr.Diethyl ether extract Ethanol extractAcetone extractGram-positive: *Bacillus cereus*, *Staphylococcus aureus*, *Bacillus subtilis*, *Streptococcus*, *Listeria monocytogenes*Gram-negative: *Escherichia coli*, *Proteus vulgaris*, *Pseudomonas aeruginosa*, *Pseudomonas syringae*, *Aeromonas hydrophila*, *Yersinia enterocolitica*, *Klebsiella pneumoniae*AntibacterialAntimicrobial activity against *B. cereus*, *S. aureus*, *E. coli*, *P. vulgaris*, *P. aeruginosa*, *Streptococcus*, *B. subtilis*, *A. hydrophila*, *L. monocytogenes*[13]Diethyl ether extractEthanol extractAcetone extract*Penicillum* sp., *Cladosporium* sp., *Fusarium oxysporum*, *F*. *culmorum*, *F*. *moniliforme*, *F*. *solani*, *Rhizopus* sp, *Aspergillus* sp.AntifungalNo antifungal activity detected[13]Acetone extractTA98 and TA100 strains of *S. typhimurium*Antigenotoxicity↑ Inhibition of frameshift mutations in TA98 than in TA100[129]Methanol extract*Salmonella typhimurium* TA1535 and TA1537
*E. coli* WP2uvrA Human lymphocyte cellsAntigenotoxicityAntimutagenic activity against *Salmonella typhimurium*No activity against *E. coli*↓ formation of SCE[14]Methanol extractEthyl acetate extractRadical scavenging activityAntioxidantMethanol extract >>> ethyl acetate extractMethanol extract: DPPH (IC_50_ 51.6 µg/mL); lipid peroxidation inhibition capacity (IC_50_ 45.5 µg/mL); ferrous ion chelating capacity (IC_50_ 50.4 µg/mL; hydroxyl radical scavenging activity (IC_50_ 90.1 µg/mL)[112]
Methanol extractHuman lymphocytes cellsAntioxidant↑ SOD, GPX, MDA levels↓ GSH[14]
Acetone extractHeLa cells, A549 cells and 5RP7 cellsCytotoxic↓ Cell viability[129]
Methanol extract*Salmonella typhimurium* TA1535
*E. coli* WP2uvrAGenotoxicityNo activity[14]*Cetraria islandica* (L.) AchMethanol extractAcetone extractLight petrolatum extractAqueous extract*Helicobacter pylori*AntibacterialLight petrolatum extract > Acetone extract[108]Methanol extractGram-positive: *Staphylococcus aureus*, *Bacillus subtilis*, *Bacillus cereus*Gram negative: *Escherichia coli*, *Proteus mirabilis*AntibacterialAntimicrobial activity against all bacteria[15]Methanol extract*Aspergillus flavus*, *Candida albicans*, *Fusarium oxysporum*, *Penicillium purpurescen*, *Trichoderma harsianum*AntifungalAntifungical activity against all fungi[15]Aqueous extractStreptozotocin-induced Diabetes Mellitus type 1 Sprague-Dawley ratsAntidiabeticSlight insulin increaseNo inhibition of glucose levels↓ infiltration of immune cells, vacuolization, and intensity of fibrosis in the kidney↑ SOD and GSH, ↓ MDA[16]Aqueous extractStreptozotocin-induced Diabetes Mellitus type 1 Sprague-Dawley ratsAntidiabeticNo body weight change↓ glucose↑ insulin levels↑ SOD, CAT and GSH levels↑ glycogen of hepatocytes↓ intensity of fibrosis[119]Aqueous ExtractStreptozotocin-induced Diabetes Mellitus type 1 Sprague-Dawley ratsAntidiabetic↓ TOS↑ TAC↑ regeneration and erythropoiesis↑ MCV, MCH, MCHC[118]*Cetraria islandica* (L.) AchAqueous extractBSA-induced arthritis in ratsAnti-inflammatory↓ reduction in the diameter between the right and left knee[90]Aqueous extractStreptozotocin-induced Diabetes Mellitus type 1 Sprague-Dawley ratsAntioxidant↑ SOD and CAT↓ MDA levelLight prevention of pancreatic cells destruction[120]Aqueous extractStreptozotocin-induced Diabetes Mellitus type 1 Sprague-Dawley ratsAntioxidant↑ SOD, GSH↓ MDA levelsPrevention of renal cell destruction.[16]Aqueous extractHuman erythrocytes with type 1 diabetes mellitusAntioxidant↑ SOD, CAT and GPx↓ MDA levels[117]Methanol extractBlood lymphocytes from human nonsmoking healthy volunteersAntioxidant↑ SOD and GPx↓ MDA[113]Aqueous extractRadical scavenging activityAntioxidant96–100% inhibition upon lipid peroxidation of linoleic acid system↑ Superoxide radical scavenging activity[114]Ethanol extractRadical scavenging activityAntioxidantDPPH, FRAP and ABTS[115]Methanol extractRadical scavenging activityAntioxidantDPPH (IC_50_ 678.3 μg/mL)Superoxide anion scavenging activity (IC_50_ 792.4 μg/mL)Reducing power range 0.0512 to 0.4562 μg/mL[15]MelaninRadical scavenging activityAntioxidantDPPH (IC_50_ 405 μg/mL)[105]Protolichesterinic acid, Lichesterinic acid, Protocetraric acid, Fumarprotocetraric acid*Trypanosoma brucei brucei*AntitrypanosomalProtolichesterinic acid MIC value 12.5 µMLichesterinic acid MIC value 6.30 µMProtocetraric acid and Fumarprotocetraric acid no antitrypanosomal activity detected[109]β-1,3/1,4-Glucan lichenanKeratinocytes (NHEK) cellsHaCaT keratinocytes cellsCellular differentiation↓ Proliferation↑ CytoKeratin 10 (CK) in the cytoplasm↑ Involucrin expressionDose-dependent CK gene expression regulation↑ Involucrin transcription levels↑ Transglutaminase gene expressionGene expression regulation of loricrin and filaggrin↑ Gene group related to cellular differentiation[97]Protolichesterinic acidA549 cellsCytotoxicNo change in 5-lipoxygenase activity↓ LRRC8A expression↓ cell viability[126]Protolichesterinic acidT-47D cells, K-562 cells andZR-75-1 cellsCytotoxicMorphological changes in T-47D and K-562↓ Cell viability↓ DNA synthesisInhibition of 5-lipoxygenase[125]Ethanol extractMCF7 cellsCytotoxic↓ Cell viability (IC_50_ 9.2047 × 10^−5^ g/mL)↓ PPAR-g levels↑ AMPK-α1 and ERK1/2 levels↑ Apoptotic cell percentage after 24 h↓ P53, Caspase 3 and Bcl-2 dose dependent[124]Methanol extractFemX and LS174 cellsCytotoxicFemX (IC_50_ 22.6 μg/mL)LS174 (IC_50_ 33.7 μg/mL)[15]*Cetraria islandica* (L.) AchLichenanU937 cellsCytotoxicNo active[128]Methanol extractMCF-7 and HepG2 cellsCytotoxicMCF-7 (IC_50_ 181.0 µg/mL)HepG2 (IC_50_ 19.5 µg/mL)[121]Fumarprotocetraric acidT-47D and Panc-1CytotoxicNo antiproliferative effect[127]Chloroform–methanol, extractDeveloping zebrafish embryosDepigmenting↓ Pigmentation (IC_50_ 44 µg/mL)[130]Chloroform–methanol extractRadical scavenging activityMeWoDepigmentingTyrosinase inhibition (IC_50_ 86 µg/mL)Cell viability assay (IC_50_ 264 µg/mL)↓ Melanin levels[130]Aqueous extractHuman erythrocytes with type 1 diabetes mellitusGenotoxicity↑ Proliferation index↓ DNA damage↓ SCE[117]Methanol extractPeripheral venous bloodGenotoxicity↑ Number of BN cells containing MNi and number of MNi in BN cells[15]Aqueous extractFumarprotocetraric acid, Protolichesterinic acid, Lichenan and isolichenanHuman monocytes differentiated into mature dendritic cells.ImmunomodulatingAqueous extract and lichenan were active↑ CD86 and ↓ CD209 and IL-12p40/IL-10[90](1 --> 3) -(1 --> 4)-α-D-Glucan polysaccharide Ci-3Whole bloodImmunomodulating↑ Granulocytic phagocytosis↓ Complementarily induced hemolysis[123]Fumarprotocetraric acidRadical scavenging activitySH-SY5Y and U373-MG cellsNeuroprotectiveORAC (5.07 μmol TE/mg), DPPH (IC_50_ 1393.83 μg/mL)↑ cell survival, GSH/GSSG↓ lipid peroxidation, ROS caspase-3 activationAvoid mitochondrial dysfunction and alterations in calcium homeostasis↓ Pro-apoptotic signalsNrf2 pathway[122]*Cetraria islandica* (L.) AchMethanol extractU373 MG cellsNeuroprotectiveORAC (3.06 µmol TE/mg), DPPH (IC_50_ 1183.55 µg/mL)↑ Cell viability and GSH/GSSG ratio↓ ROS generation and lipid peroxidation[121]*Cetraria pinastri* (Scop.) Gray.Methanol extractGram-positive: *Enterococcus fecalis*, *Staphylococcus aureus*.Gram-negative: *Escherichia coli*, *Klebsiella pneumoniae Micrococcus lysodeikticus*, *Pseudomonas aeruginosa*AntibacterialAntimicrobial activity against all bacterial strains[107]*Cetraria pinastri* (Scop.) Gray.Methanol extract*Alternaria alternate*, *Aspergillus flavus*, *A. niger*, *Candida albicans*, *Cladosporium cladosporioides*, *Paecilomyces variotii*, *Acremonium chrysogenum*, *Fusarium oxysporum*, *Penicillium verrucosum Trichoderma harsianum*AntifungalAntifungal activity against all fungal species tested[107]*Cetraria pinastri* (Scop.) Gray.Methanol extractThiocyanate methodAntioxidant48.79% inhibition of the oxidation of linoleic acid[11]


## 7. Conclusions and Future Prospects

Within the family of Parmeliaceae, the genus *Cetraria* s. str. is one of the least investigated groups in terms of certain areas of knowledge, such as pharmacology and ecology/environmental interest, and is one of the most complex genera in terms of taxonomy/phylogeny, as shown by the discrepancy in terms of classification. It is noteworthy that most studies focused on *Cetraria islandica*, with the rest of the species of this genus being very sparsely studied in all the aspects included in this review. Further, pharmacological investigations that validate traditional uses and discover new activities are very limited. Most have been carried out with extracts, and very few with isolated compounds. Furthermore, these pharmacological experiments have been conducted mainly in vitro, and few have been in vivo. Likewise, to date, there are no clinical trials using species of the genus *Cetraria* to evaluate safety, efficacy, and toxicity. Given the lack of focus on the therapeutic potential of *Cetraria* species, it would be necessary to perform more studies to evaluate new activities, discover novel bioactive compounds and elucidate the mechanisms of action. Our study opens a new window to evaluate the therapeutic potential of *Cetraria* species, and lichens in general, as a source of new drugs.

## Figures and Tables

**Figure 1 molecules-27-04990-f001:**
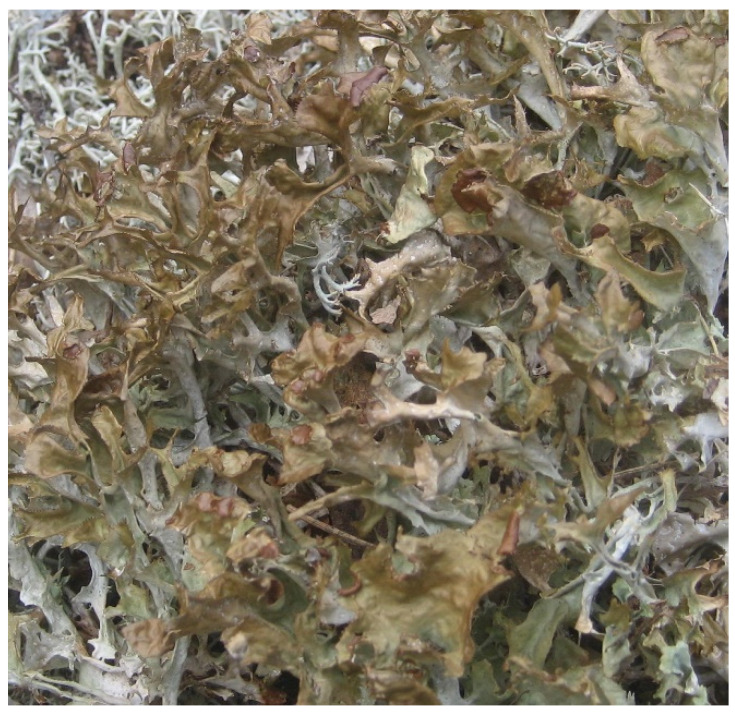
*Cetraria islandica* (L.) Ach. Photograph was kindly provided by prof. Divakar.

**Figure 2 molecules-27-04990-f002:**
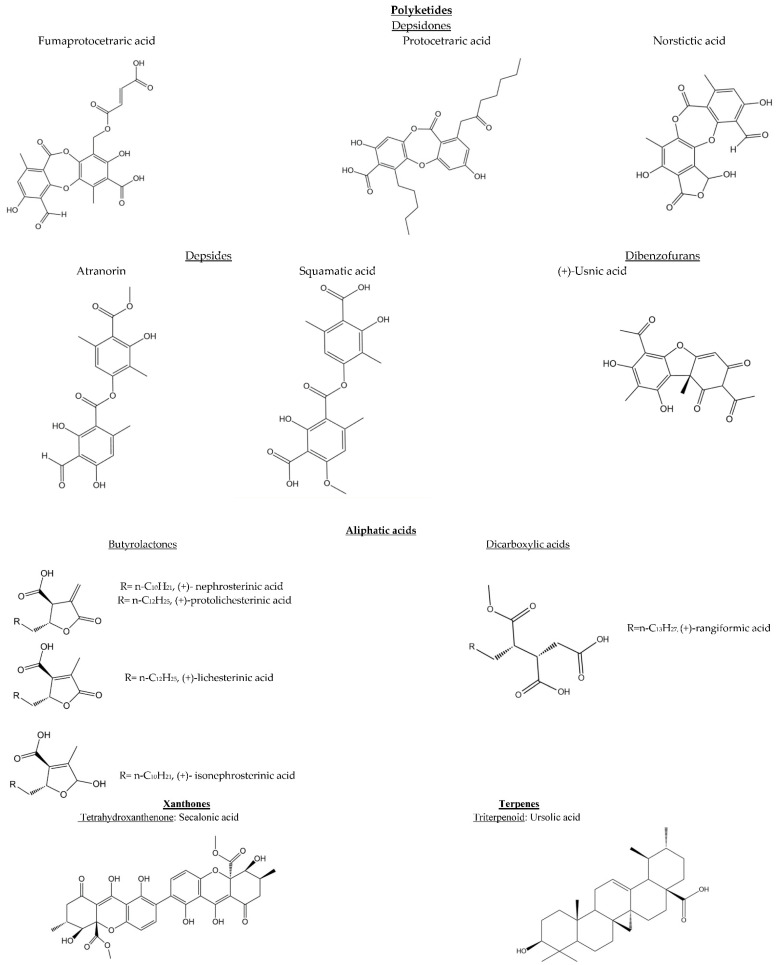
Chemical structures of *Cetraria* species secondary metabolites.

**Table 2 molecules-27-04990-t002:** Distribution patterns of *Cetraria* lichen species.

Lichen Species	Distribution Pattern	Distribution Areas	References
*Cetraria aculeata* (Schreb.) Fr.	Cosmopolitan	Four continents and many oceanic islands.	[41]
*Cetraria annae* Oxner	Endemic	Baikal region, Russia	[30]
*Cetraria australiensis* W.A.Weber ex Kärnefelt	Endemic	South-eastern Australia	[41]
*Cetraria crespoae* (Barreno & Vázquez) Kärnefelt	Endemic	Western parts of the Iberian Peninsula and Italy	[29]
*Cetraria ericetorum* Opiz Sp *ericetorum* Sp *reticulata* Sp *patagonica*	Endemic (each subspecies separated geographically)	Europe and Asia North America Southern part of South America	[36,45]
*Cetraria islandica* (L.) Ach. Sp *islandica* Sp *crispiformis* Sp *antartica*	Bipolar Circumboreal Austral	High latitudes in northern and southern hemispheres Europe, Asia, North America Southern hemisphere	[41,45]
*Cetraria kamczatica* Savicz	Amphi-Beringian	Eastern Siberia and Alaska	[37]
*Cetraria laevigata* Rass.	Amphi-Beringian	North America (from Alaska through upper Canada)	[44,45]
*Cetraria minuscula* (Elenkin and Savicz) McCune	Amphi-Beringian	Eastern Siberia, Interior Alaska	[43]
*Cetraria muricata* (Ach.) Eckfeldt	Cosmopolitan	Four continents and many oceanic islands.	[41]
*Cetraria nepalensis* D.D. Awasthi	Endemic	Isolated localities at high elevations in the Great Himalayas	[45]
*Cetraria nigricans* Nyl.	Circumpolar	High Arctic (Alaska, Canada) Alpine sites in southern areas	[41,44]
*Cetraria odontella* (Ach.) Ach.	Circumboreal	Boreal regions of the northern hemisphere.	[41]
*Cetraria obstusata* (Schaer.) Van den Boom & Sipman	Endemic	Alpine areas of the Alps—Austria, Italy, Switzerland	[34]
*Cetraria peruviana* Kärnefelt & Thell	Endemic	Central part of South America	[41]
*Cetraria rassadinae* Makryi	Endemic	Northern Baikal region of central Siberia	[45]
*Cetraria sepincola* (Hoffm.) Ach.	Circumboreal	Boreal forest and in the tundras of the Arctic	[41,44]
*Cetraria steppae* (Savicz) Kärnefelt	Endemic	Semiarid Eurasian steppe biomes from Kazakhstan to Iran and Ukraine.	[35,45]

**Table 3 molecules-27-04990-t003:** Chemical composition of *Cetraria* spp.

Lichen Species	Chemical Composition	References
*Cetraria aculeata* (Schreb.) Fr.	Lichesterinic acid, protolichesterinic acid	[34]
*Cetraria annae* Oxner	Major: usnic acid, isonephrosterinic acid Minor: lichesterinic acid, atranorin, squamatic acid Trace: protolichesterinic acid, nephrosterinic acid	[30]
*Cetraria australiensis* W.A.Weber ex Kärnefelt	Lichesterinic acid, protolichesterinic acid, ursolic acid.	[30]
*Cetraria ericetorum* Opiz	Lichesterinic protolichesterinic acid	[33]
*Cetraria islandica* (L.) Ach.	Fumarprotocetraric acid, protocetraric acid, protolichesterinic acid, usnic acid	[12]
*Cetraria kamczatica* Savicz	Protolichesterinic acid, rangiformic acid	[44]
*Cetraria laevigata* Rass.	Fumarprotocetraric acid.	[44]
*Cetraria muricata* (Ach.) Eckfeldt	Lichesterinic acid, protolichesterinic acid	[30]
*Cetraria nigricans* Nyl.	Protolichesterinic acid, rangiformic acid, secalonic acid	[34]
*Cetraria odontella* (Ach.) Ach.	Protolichesteric and rangiformic acids	
*Cetraria obstusata* (*Schaer*.) Van den Boom and Sipman	Lichesterinic acid, protolichesterinic acid, secalonic acid	[34]
*Cetraria sepincola* (Hoffm.) Ach.	Lichesterinic acid and protolichesterinic acid.	[34]
*Cetraria steppae* (Savicz) Kärnefelt	Usnic acid, lichesterinic acid, protolichesterinic acid, norstictic acid	[35]

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
