# Peer review of "The Genus Cetraria s. str.—A Review of Its Botany, Phytochemistry, Traditional Uses and Pharmacology"

_molecules, 2022, doi:10.3390/molecules27154990_

Round 1

Reviewer 1 Report

   It is not a thorough review of  the botany, phytochemistry and pharmacology of Cetraria. There are important literatures missed, such as chemistry of Cetraria islandica, Rassabina et al, 2020, Biochemistry-Moscow, Zarchaski et al, 2018, Fitotrapia, and pharmacology of it, Chang et al, 2021, Cell, Thorsteinsdottir et al, 2016, Phytotherapy Research and Johnnsson et al, 2021, Planta Medica.

   The author list seems has not finished. The chemical structure graphs need to be unified in same size and drawing style. Organism's scientific names in text and literature lists need to be italic.

Reviewer 2 Report

We can accept this paper with some minor revisions like the chemical structures are not managed properly in the paper. Second comment I can suggest here is to provide the compounds based on their chemical classification and provide the structures of most active or bioactive compounds in the paper.

1. Corrrect figure 2, all the molecules are overlapping. There is sufficient space to rearrange the structures.

2. It would have been better if the author can provide the chemical constituents based on there chemical classes with the structures of some majour bioactive compounds from genus Cetraria s. str.

Reviewer 3 Report

This review describes many activities of lichen extracts, but not at the molecular level. The various activities are explained mostly for the extract. Therefore, due to poor chemistry, I recommend submission to some other journal. 

Other comments

1. Carotenoids are not primary but secondary metabolites (page 10, line 2 from the bottom).

2. Please see any chemistry journal for the arrangement of compounds (Fig. 2). 

3. Configuration of each compound should be drawn (Fig. 2). 

4. Some abbreviations in this manuscript are not common in chemistry. Please define them when first mentioned. 

Round 2

Reviewer 1 Report

I did not ask to review this manuscript again. It is not a good review still,  serious mistakes remain. 

Reviewer 3 Report

I did not review due to computer problem. 
